# Predicting Breast Cancer Events in Ductal Carcinoma In Situ (DCIS) Using Generative Adversarial Network Augmented Deep Learning Model

**DOI:** 10.3390/cancers15071922

**Published:** 2023-03-23

**Authors:** Soumya Ghose, Sanghee Cho, Fiona Ginty, Elizabeth McDonough, Cynthia Davis, Zhanpan Zhang, Jhimli Mitra, Adrian L. Harris, Aye Aye Thike, Puay Hoon Tan, Yesim Gökmen-Polar, Sunil S. Badve

**Affiliations:** 1GE Research Center, Niskayuna, NY 12309, USA; 2Department of Oncology, Cancer and Haematology Centre, Oxford University, Oxford OX3 9DU, UK; 3Anatomical Pathology, Singapore General Hospital, Singapore 169608, Singapore; 4Department of Pathology and Laboratory Medicine, Emory University School of Medicine, Atlanta, GA 30322, USA; ypolar@emory.edu; 5Winship Cancer Institute, Atlanta, GA 30322, USA

**Keywords:** DCIS, breast cancer events, risk-stratification, deep learning

## Abstract

**Simple Summary:**

Ductal carcinoma in situ (DCIS) patients have an excellent overall survival rate and over-treatment is always a cause for concern due to potential side-effects. Standard clinicopathological parameters have limited value in predicting breast cancer events (BCEs) and stratification of high and low risk patients. Herein, we have developed a deep learning (DL) classification framework to predict BCEs in DCIS patients. A generative adversarial network (GAN) augmented deep learning (DL) classification of histological features associated with aggressive disease was trained on hematoxylin and eosin (H & E) tissue microarray (TMA) images of DCIS to predict BCEs. The area under the curve (AUC) for BCE’s in the validation set was 0.82. Early and accurate prediction of DCIS BCEs would facilitate a personalized approach to therapy.

**Abstract:**

Standard clinicopathological parameters (age, growth pattern, tumor size, margin status, and grade) have been shown to have limited value in predicting recurrence in ductal carcinoma in situ (DCIS) patients. Early and accurate recurrence prediction would facilitate a more aggressive treatment policy for high-risk patients (mastectomy or adjuvant radiation therapy), and simultaneously reduce over-treatment of low-risk patients. Generative adversarial networks (GAN) are a class of DL models in which two adversarial neural networks, generator and discriminator, compete with each other to generate high quality images. In this work, we have developed a deep learning (DL) classification network that predicts breast cancer events (BCEs) in DCIS patients using hematoxylin and eosin (H & E) images. The DL classification model was trained on 67 patients using image patches from the actual DCIS cores and GAN generated image patches to predict breast cancer events (BCEs). The hold-out validation dataset (*n* = 66) had an AUC of 0.82. Bayesian analysis further confirmed the independence of the model from classical clinicopathological parameters. DL models of H & E images may be used as a risk stratification strategy for DCIS patients to personalize therapy.

## 1. Introduction

Ductal carcinoma in situ (DCIS) accounts for 20–25% of the cases of breast cancer [1,2] and is estimated to have afflicted over 1 million US women in 2022. Patients with DCIS are at risk of developing recurrent DCIS or invasive carcinoma. The incidence of ipsilateral disease breast cancer events (BCEs) is approximately 30% in patients treated by surgical excision [3]. Clinical trials have documented that this risk for recurrent disease can be reduced by approximately 50% with the addition of radiation [3]. Mastectomies or surgical excisions plus radiation have been adopted as the standard of care for the management of DCIS [3]. DCIS patients have an excellent overall survival rate and over-treatment is always a cause for concern due to potential side-effects [4,5,6]. Standard clinicopathological factors (age, growth pattern, tumor size, margin status, and grade) have been shown to have limited value in predicting development of additional BCEs and segregation of high and low risk patients [7]. Molecular prognostic variables of DCIS have not offered consistent results nor have they been found to be of prognostic value [8]. mRNA-based models have been described [9,10,11] but have not been considered to be cost-effective [12]. Thus, there is an unmet clinical need for novel tools to improve BCEs risk stratification in DCIS [1,2,13].

In the last decade, digitized hematoxylin and eosin (H & E) images have been increasingly used in machine learning frameworks for prognosis prediction in a variety of cancer types. The extracted morphometric and texture feature images have been used in a classification framework [14,15,16]. Histologic parameters such as nuclei shape, spatial arrangements, and necrosis within ducts have also been shown to outperform traditional clinicopathological variables in predicting BCEs in DCIS [17,18]. However, these models are narrowly focused on specific DCIS characteristics and discard rich information that could be derived from the tumor microenvironment including surrounding blood vessels, stromal characteristics and distribution, intra- and inter-tumor heterogeneity and their combinations. In more targeted feature analysis, we, amongst others, have documented a prognostic role of tumor infiltrating lymphocytes in DCIS [19,20,21,22].

In recent years, deep convolutional neural networks (CNN) trained on H & E images have been increasingly used for cancer prediction [23,24]. Unlike traditional machine learning models that use hand crafted morphometric and texture features, these models automatically learn the prognostic features from a large number of manually annotated images. Large deep CNNs or DL networks that have been trained with natural images for classification are re-tuned with a small number of manually annotated H & E images in a transfer learning framework for prognosis prediction. During the transfer learning process, the last few layers of a large DL network is re-trained using the small set of manually annotated images. Performance of such a transfer learning network is heavily dependent on (a) data volume, (b) data variability, and (c) the ratio of benign to malignant images, often referred to as class balance. Digitized H & E images often suffer from a lack of data variability and a class imbalance problem. For example, if large regions of the images are normal, it can introduce a class imbalance issue for model training. Manually annotated (or automatically detected) cancer regions could potentially be used for training, however this can reduce data volume and variability. Data variability and an imbalanced dataset can adversely affect transfer learning of DL networks, resulting in poor performance. In such situations, traditional machine learning models based on hand-crafted features may outperform DL-networks [14].

Generative adversarial networks (GANs) are a class of DL models in which two adversarial neural networks, generator and discriminator, compete with each other to generate high quality images. During the adversarial training process, the generator learns to synthesize realistic images similar to those in the training set. The discriminator learns to distinguish between real and generated images. In recent years, GAN has been increasingly used for digital pathology and histological image analysis including image processing and data augmentation, where high quality H & E images have been generated [25,26,27,28,29]. The pathology GAN (P-GAN) image patch generation model has been used to generate high fidelity H & E images that capture key tissue features that represent aggressive cancer characteristics [30]. P-GAN, however, is restricted to image generation and does not allow prognosis prediction. We have modified P-GAN to develop a novel lethality-GAN (L-GAN) augmented classification model that generates aggressive images with features that are similar to clinical images from patients with poor prognosis as shown in Figure 1. These images were used to augment the clinical dataset to introduce data variability and improve the performance of the DL classification model (Inception-Resnetv2).

The generated patches were encoded with a DL network Inception v3 and projected to the feature space. Similarly, patches from “true” H & E images were encoded and projected into the feature space. GAN generated patches with close proximity to aggressive cancer patches from clinical H & E images in the feature space are selected to train the DL classification model. The ability to automatically generate infinite aggressive malignant H & E image patches alleviates the issues that are related to data variability and class imbalance that are often encountered in re-tuning a large DL network. The re-tuned DL network classifier was then used to predict BCE in an independent cohort. To the best of our knowledge, this is the first time a GAN model has been used to generate aggressive malignant H & E image patches to train a DL classification model to predict BCEs in DCIS patients.

## 2. Materials and Methods

### 2.1. Clinical and Histological Data

After the obtaining a waiver of consent requirement for non-human subjects’ research from Indiana University institutional review board (IRB), histologically confirmed cases of DCIS were obtained from the pathology databases of Oxford University (Oxford, UK), Singapore General Hospital, (SGH) Singapore. All cases had to have either a history of development of a BCE or a minimum of 3 years follow-up without any additional breast event. BCE, for the purposes of the study, was defined as any (ipsilateral or contralateral) in situ or invasive (local or distant) breast cancer. All cases were de-identified before analyses. Cases of papillary DCIS were excluded from analysis. A multi-institutional cohort consisted of 133 cases of which 45 cases had BCEs; detailed descriptions of both the cohorts have been provided in Table 1. In brief, the Oxford or the training cohort comprised of 67 DCIS patients aged between 32 and 75 years, with a lesion size of 19 mm on average, with a mix of low (13%), intermediate (29%), and high grade (57%) cases. TMAs were constructed from 2 mm cores in conjunction with a breast pathologist and 1–3 cores per patient were available for H & E image analysis. The validation dataset, Singapore, cohort comprised of 66 DCIS patients, aged between 35 and 80 years, a lesion size of 5–90 mm, with a mix of low (12%), intermediate (33%), and high grade (55%) cases. TMA cores were 2 mm with 3 cores per patient. Patients from both the cohorts were treated with a mastectomy and/or a combination of lumpectomy, radiation, and hormone therapy.

### 2.2. GAN Augmented DL Classification Model

The H & E image of each TMA core was approximately 7000 × 6000 pixels. A 2-class Gaussian mixture modeling approach was applied to each of the cores to separate tissue class versus background. A high-quality TMA core (i.e., no tears, intact) was manually chosen as a reference template. All other cores were normalized to the reference template using histogram equalization with 128 bins across RGB color channels. Patches with 256 × 256 pixels were extracted from each tissue core post histogram equalization for DL training and validation. Patches with less than 50% of area with tissue were automatically discarded from analysis.

Pathology GAN (P-GAN) [30] has been employed to generate high fidelity H & E images that capture key tissue features representing aggressive cancer characteristics. P-GAN, however, is restricted to image generation and does not allow prognosis prediction. Feature space of P-GAN created from image patches from H & E image shows pathologically meaningful representation of lethal cancers clustered together in the embedded feature space. In our novel lethality-GAN (L-GAN) augmented classification model, Figure 1, we have modified P-GAN to generate aggressive H & E images with features that are similar to images with poor prognosis. We use these to augment the dataset to improve data variability and improve the DL classification model (Inception-Resnetv2).

While the P-GAN [25] network is capable of generating high fidelity image patches with key tissue features such as color, texture, shape, and spatial arrangement of both normal and cancer cells, the L-GAN framework is specifically capable of capturing the aggressive image patches that are generated using the underlying P-GAN [25]. The DL encoder/mapping network (Inceptionv3) within the GAN framework was re-tuned using a transfer learning strategy to learn the latent feature space of BCEs in DCIS patches that comprised of embeddings of color, shape, texture, and spatial features. The re-tuned encoding network was then used to extract features of the GAN-generated image patches. The generated image patches that were similar to original aggressive image patches in feature space were automatically selected based on cosine similarity [31] or cosine angular similarity of the feature vectors. Cosine similarity of feature vectors that were extracted by the encoding/mapping network (from aggressive cancer patches and GAN-generated image patches) were used to identify aggressive image patches that were generated by GAN. The selected aggressive image patches that were generated by GAN were then used to augment data for the subsequent DL classification model (Inception-Resnetv2) training for BCE prediction in DCIS patients.

The ability to generate such infinite aggressive image patches permits control of data variability, reduce class imbalance (aggressive versus less aggressive patches), and generates a large dataset for transfer learning from a smaller cohort of cases. Retuning/re-training more layers of the Inception-Resnetv2 network with a balanced dataset resulted in significant improvement in sensitivity and specificity compared to training with real H & E image patches which are not balanced. Given a new H & E test image, overlapping (50% overlap) image patches were extracted and classified as aggressive/lethal or non-aggressive/non-lethal by the re-trained Inception-Resnetv2 DL network. The ratio of aggressive to non-aggressive patches for a DCIS case was then used as a threshold to classify patient risk. The ratio was automatically determined from a set of validation images from the training cohort and applied to the hold out test cohort.

### 2.3. Bayesian Network Analysis of BCE Risk Factors

Bayesian network analysis was used to study the association our L-GAN model with BCE, incorporating the clinicopathological data and tumor infiltrating lymphocytes (which were generated in an earlier published study using the same cohorts [22]). The tumor infiltrating lymphocyte (TIL) parameters included stromal TILs, touching TILs, circumferential TILs, and hotspots which were manually scored by a pathologist using the same H & E images as the current study. A TILs risk score was trained on the training (Oxford) data using a Cox-proportional hazard model and applied to the validation (Singapore) cohort.

A Bayesian network is characterized by a directed acyclic graph consisting of a set of nodes (variables) and the direct and indirect dependencies among the nodes [32]. The Bayesian network structure is learned on the Singapore cohort by the hill-climbing algorithm with the Akaike Information Criterion score given the constraints that “Age” can only be a parent node and “Breast Cancer Event” can only be a child node. We evaluated the performance of the Bayesian network model using the leave-one-out validation method. The advantage of a Bayesian network approach over other multivariable regression model approaches is that it facilitates multi-modal fusion and integration in an easy-to-interpret manner. Conditional probabilities derived from a Bayesian network also allow investigation of connections between the clinicopathological variables and diagnostic parameters for an individual patient providing interpretable personalized BCE risk score. We also explored pairwise correlation among all the factors to understand pairwise associations prior to integrating all the factors with BCE. The dependence of each pair was evaluated by Fishers exact test, and adjusted *p* (FDR) = 0.1 was applied to 10 pairs; *p*-value was adjusted using Benjamini–Hochberg method.

## 3. Results

### 3.1. DL Network Performs Significantly Better when Augmented with L-GAN-Generated Aggressive Cancer Image Patches

The L-GAN model was first trained with the Oxford cohort (*n* = 67). During GAN training, the Oxford cohort was randomly partitioned into an 80% training and 20% validation split to create the generative model which automatically generated H & E image patches. Aggressive image patches (*n* = 300,000) were automatically generated to train the Inception-Resnetv2 classification model. The use of over 300,000 P-GAN-generated image patches yielded similar results.

Inception-Resnetv2 (Figure 1) was selected as the classification model as it has shown superior performances in natural images as compared to prior DL classification networks such as ResNet50, Inceptionv2, and Inceptionv3 [33]. During re-tuning of the Inception-Resnetv2, the Oxford cohort was randomly split into 90% for training and 10% for testing. All 300,000 aggressive image patches that were generated by the GAN network were used for re-tuning the Inception-Resnetv2 classification network to predict BCEs. We created three models progressively re-tuning the last 20, 52, and 84 layers obtaining an AUC of 0.86, 0.89, and 0.87. The optimal performance of AUC 0.89 was obtained with re-tuning of last 52 layers; this was selected for further validation. The re-tuned Inception-Resnetv2 classification model was validated using the Singapore cohort (*n* = 66). Application of the model to the validation cohort resulted in a mean AUC of 0.82 for predicting BCEs with a sensitivity of 0.81 and specificity of 0.83.

The AUC of the (un-modified) Inception-Resnetv2 classification model when trained with real H & E images was 0.82 for the Oxford cohort and 0.77 for the Singapore cohort. Thus, the use of the L-GAN led to a 7% improvement in performance images in the validation cohort.

### 3.2. Correlation of L-GAN and Clinicopathological Data

We next analyzed the pairwise correlation of L-GAN and TILs models with clinicopathological parameters including age, size, and grade using Fisher’s exact test. As shown in Figure 2, age and grade were significantly correlated with all the risk factors except L-GAN. Grade demonstrated a positive correlation with age (older patients tended to have higher grade; adj.*p* = 0.03), with tumor size (adj.*p* < 0.001), and TILs risk model (adj.*p* = 0.01).

### 3.3. Bayesian Network Provides a Scalable Interpretable Framework for Combining Multi-Modal Risk Score

For the Bayesian network structure (Figure 3), we estimated the parameters or the conditional probabilities of the local distributions for each node which describes the relationship between the parent node(s) and itself. For the “size” node, its local distribution is given by a 3-by-2 conditional probability table (Figure 3), each row of which is for one level of its parent node (“Grade”). For example, for Grade = High, the probability of size > 20 mm is 72%. Likewise, for the “Breast Cancer Event” node, its local distribution is given by a 4-by-2 conditional probability table, each row of which is represents one combination of levels of its parent nodes (“Age” and “GAN”). With the learned Bayesian network structure and estimated local distributions, the probability of BCE can be inferred based on knowledge of associated factors. The integrated analysis provided an AUC of 0.843 using the leave-one-out cross validation, while each of the single modalities had an AUC ranging from 0.52 to 0.82. The Kaplan–Meier analysis showed the risk of BCE between the high versus low risk groups (Figure 4) was statistically significant (*p* < 0.001).

## 4. Discussion

The combination therapy of surgery, radiation, and endocrine therapy reduces the risk of BCEs in patients that were diagnosed with DCIS to around 10% [3], which has resulted in all patients with DCIS being treated with all three regimens. However, almost 70% of DCIS patients that were treated with surgery alone will not develop another breast cancer event (BCE) [3]. This highlights the need for a predictive assay to identify high risk patients and reduce over-treatment in low risk patients. We have previously developed, in collaboration with Genomic Health Inc (now Exact Sciences), a gene expression assay (DCIS score) for the prediction of BCE risk in DCIS [11]. However, this assay is relatively expensive and has not been widely used [12]. Furthermore, the inherent histological heterogeneity of DCIS means that bulk genomic data may be diluted/contaminated with normal regions and using histological images may be more clinically relevant. Partially addressing this, Klimov et al. [13] used a two-step approach to develop a predictive algorithm using H & E images. In the first step, a classifier that was trained manually by pathologists was applied to H & E images to annotate the areas of stroma, normal/benign ducts, cancer ducts, dense lymphocyte region, and blood vessels. In the second step, a BCE risk classifier was trained on eight selected architectural and spatial organization tissue features from the annotated areas to predict BCE risk. The developed eight-feature classifier could significantly predict BCE risk. However, manual annotation is time consuming, expensive, and suffers from inter- and intra-rater variabilities.

Our group has previously attempted to develop an AI-based classifier of DCIS using classical DL tools [18]. In a study by Li et al. [18], we evaluated the ability of quantitative nuclear histomorphometric features that were extracted from H & E-stained slide images of 62 ductal carcinoma in situ (DCIS) patients to distinguish between DCIS score risk categories [18]. Features relating to spatial distribution, orientation disorder, and texture of nuclei were identified as most discriminating between the high DCIS score and the intermediate-low DCIS score risk categories. However, in spite of using the most discriminating set of features, the AUC range that could be achieved was only in the range of 0.57 to 0.68. This highlights the need for the development of better AI-tools.

Classic DL methods require large manually annotated cohorts, which have been difficult to collect in DCIS as it is expensive, time consuming, and disrupts the clinical workflow [23]. P-GAN has been previously used for the generation of synthetic images and augmentation of prostate datasets [30]. Herein, we have adopted the GAN framework to select DCIS H & E image patches with aggressive cancer characteristics, based on their proximity to similar aggressive cancer patches in the feature space of the GAN model (L-GAN). This results in an ability to automatically generate infinite aggressive H & E image patches and resolves the issue of data variability and class imbalance issues that often handicap re-tuning of a large DL network. We further developed a risk score using this re-tuned L-GAN model and integrated it with clinicopathological data and previously derived TILs risk scores [22] using a Bayesian network. We hypothesized that a greater number of aggressive patches in a given patient would be associated with greater risk of BCEs. The results in the validation set confirming the ability of L-GAN in distinguishing with and without BCEs (AUC = 0.82) with a high sensitivity (0.81) and specificity (0.83).

Pairwise correlation identified that L-GAN is not statistically correlated with any of the risk factors. Importantly, a Bayesian network identified L-GAN and age as most closely associated with BCEs with an AUC = 0.843 (LOOCV). This is similar to the data observed in patients with invasive breast cancer enrolled in the TAILORx and RxPonder clinical trials [34,35]. In these trials, premenopausal women (defined as age less than 50 yrs) with a low or intermediate (Oncotype Dx) recurrence score benefited from chemotherapy whereas post-menopausal women did not. The Bayesian analysis led to the recognition of the importance of age as a significant variable in the prediction of BCEs.

Although the framework for DCIS BCE prediction was validated in a holdout dataset, there still some limitations to the study. The overall cohorts are small and the patients were treated in a non-homogenous manner. Surgical treatment in the form of mastectomy was routinely used in the Singapore cohort while lumpectomy was most often used in the Oxford cohort. Furthermore, there was almost no usage of endocrine therapy in the Singapore cohort. Due to the older nature of the Oxford cohort, endocrine therapy had been prescribed in only a small percentage of patients. BCE was defined as any breast cancer-related event including local and distant disease; this was in part due to the small cohort sizes and low incidence of events in patients with DCIS. In the cohorts that were used, the follow-up non-BCE patients was limited resulting in the use of a 3-year endpoint. Although the cohorts were small, the multi-center nature of the study is major strength. Further validations of L-GAN are necessary prior to large-scale clinical implementation.

## 5. Conclusions

The major contributions of our work are: (a) The use of GAN to generate a large volume of aggressive cancer H & E image patches to resolve limited data and class imbalance issue; (b) the use of feature/latent space of GAN in selection of aggressive H & E image patches to resolve data variability issue; (c) the improved performance of a large DL network using GAN-generated and real H & E image patches compared to network trained with real H & E image patches only; and (d) Bayesian network integrates various modalities into an interpretable personalized BCE risk score. The current study is a proof-of-principle study that establishes the potential of using GAN models for prediction. Further studies are necessary to assess the impact of these predictive models on clinical practice.

## Figures and Tables

**Figure 1 cancers-15-01922-f001:**
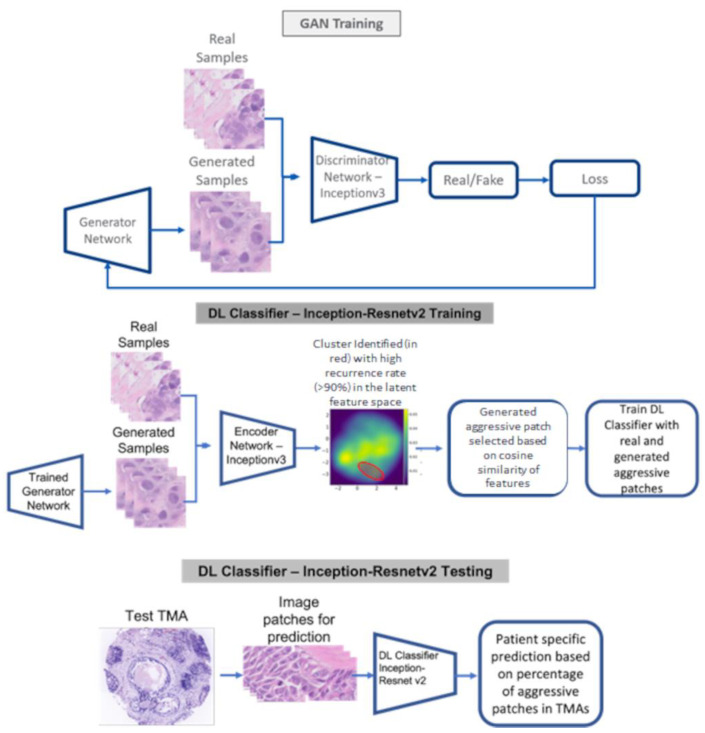
The architecture of the generative adversarial network (GAN) and its modification to develop the L-GAN framework. During GAN training, a generative model was developed that automatically generates high quality H & E image patches. The generated image patches that are similar to original aggressive image patches in latent feature space cluster were automatically selected based on the cosine similarity of the feature vectors; During DL classifier training (Inception-Resnetv2), real H & E image patches along with generated aggressive image patches were used. The trained DL classifier was then used to predict aggressiveness of a new image patches.

**Figure 2 cancers-15-01922-f002:**
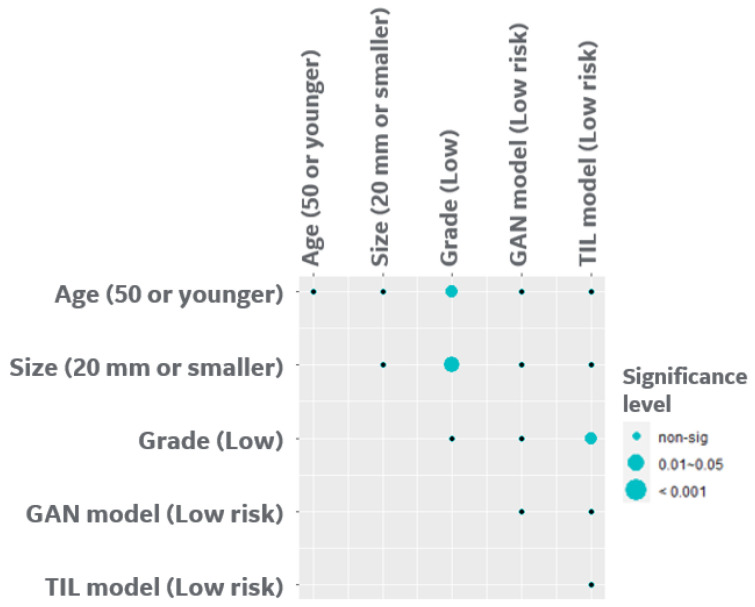
Correlation analysis of clinical factors along with risk prediction from each modality for the Singapore Cohort. The dependence of each pair was evaluated by Fishers exact test, and adjusted *p* (FDR) = 0.1 was applied to 10 pairs; *p*-value was adjusted using Benjamini–Hochberg method.

**Figure 3 cancers-15-01922-f003:**
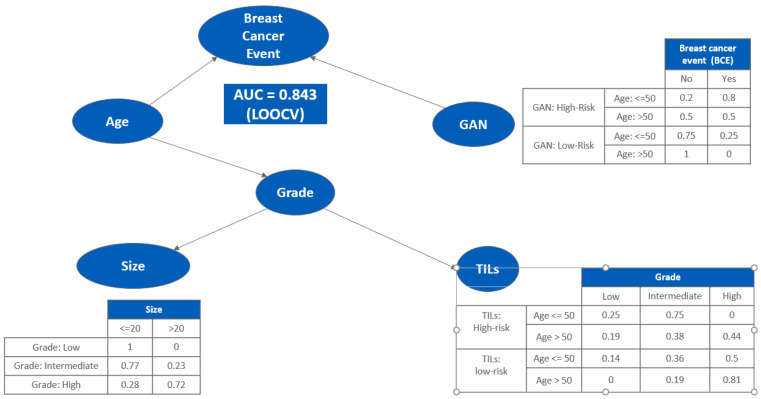
Bayesian network integrating multi-modalities with clinicopathological data and likelihood of BCEs.

**Figure 4 cancers-15-01922-f004:**
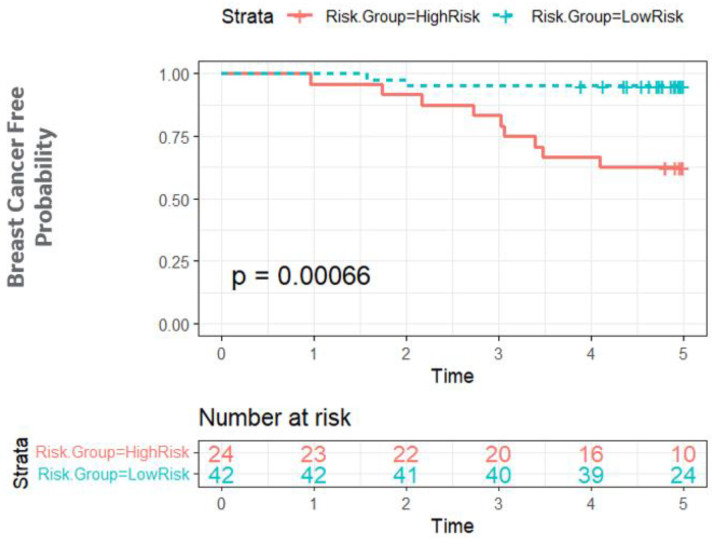
Kaplan–Meier plot on the integrated prediction of breast cancer event in the Singapore cohort (validation dataset). We divided the data into two groups based on the prediction from the Bayesian network (LOOCV prediction) and evaluated breast cancer events over 5 years.

**Table 1 cancers-15-01922-t001:** Patient Characteristics for each cohort.

	Oxford (N = 67)	Singapore (N = 66)
Follow up time		
Mean (SD)	7.74 (4.36)	5.1 (2.17)
Range (BCE)	0.83–15.17	0.96–13.71
Range (Non-BCE)	4.9–17.08	3.89–6.16
Breast Cancer Event		
N	40 (59.7%)	48 (72.7%)
Y	27 (40.3%)	18 (27.3%)
Age		
N-Miss	1	0
≤50	19 (28.8%)	23 (34.8%)
>50	47 (71.2%)	43 (65.2%)
Size		
N-Miss	39	0
≤20	18 (64.3%)	34 (51.5%)
>20	10 (35.7%)	32 (48.5%)
Grade		
N-Miss	8	0
Low	8 (13.6%)	8 (12.1%)
Intermediate	17 (28.8%)	22 (33.3%)
High	34 (57.6%)	36 (54.5%)
Lymphocyte		
0–5%	27 (40.3%)	22 (33.3%)
>5%	40 (59.7%)	44 (66.7%)
Touching TILs		
0	52 (77.6%)	52 (78.8%)
>0	15 (22.4%)	14 (21.2%)
Circumferential TILs		
No	51 (76.1%)	38 (57.6%)
Yes	16 (23.9%)	28 (42.4%)
Hotspot		
No	40 (59.7%)	30 (45.5%)
dense	27 (40.3%)	36 (54.5%)

## Data Availability

We will be happy to share the datasets and the code upon request with appropriate material transfer agreements.

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
