# Peer review of "Predicting Breast Cancer Events in Ductal Carcinoma In Situ (DCIS) Using Generative Adversarial Network Augmented Deep Learning Model"

_cancers, 2023, doi:10.3390/cancers15071922_

Round 1

Reviewer 1 Report

Starting from the assumption that there are no standard clinic-pathological factors considered accurate in predicting recurrence of DCIS and that new findings highlight possible contribution of AI to investigate cancer prognosis prediction, the authors developed a GAN augmented deep learning (DL) classification framework using haematoxylin and eosin (H&E) images for predicting recurrence in DCIS patients. The DL model was mainly used to classify patients’ risk based on the differences identified between aggressive and non-aggressive images. They also studied the relationship between GAN classification and classical clinic-pathological parameters. The purpose of the study is original.

1. Acronyms aren’t coherent with their order in text (for example H&E at line 56): they appear in text first without explanation. In general, there’s no explanation of BCE acronym.

  1. The quality of exposure is not always precise and there are some typos and sloppy mistakes. Some sentences are often not clear and too long. Grammar should be reviewed as the verb tenses are often contrasting.
  2. Author should report identification code of the protocol and institutional approval of their study
  3. Authors should explain better which kind of recurrence they have considered: local, loco-regional, ipsilateral…
  4. How many mastectomies/lumpectomies in the cohort? What about resection margins in conservative surgery? Which margins was considered as free from tumour?
  5. Did authors consider distant metastases too? If not, why not?
  6. Authors referred to a publication (reference n°22) for cohorts’ description but some questions have to be clarified:

·        Singapore cohort presented n=73 patients in reference n°22 but in this manuscript patients from the same cohort are 65. Authors should explain the reason of the exclusion and fill a table in text with baseline distribution of the new cohorts.

·        Authors wrote “All cases had to have either a history of development of a second breast cancer event or a minimum of 3 years without any additional breast event” but from Supplementary Table S1 of reference n°22, follow up time (years) of non-recurrent patients starts respectively from 0.03 and 0.9 years for Oxford and Singapore cohort. Authors should explain why they included patients without a recurrence history and without a minimum of 3 years without any additional breast event. On the other hand, if they weren’t to be included, authors should consider reperform analysis with the corrected sample.

·        Authors should explain why they consider only a minimum of 3 years without recurrence: why not 5 years for example? Some references need to be addressed for this choice, this issue could be a selection bias. How long was the follow-up period the authors considered for recurrence diagnosis? 

8. In material and method section there isn’t any part related to how researchers collected H&E images, their quality, their dimension, their format, if they were elaborated before they were given as input to L-GAN, which part was considered... Author should add this part to the section

9. Some of the basic notions about DL and GAN should be better explained, in order to be understood by a wider audience. For example: latent feature space, cosine similarity.

10. Where was reported Table 1?

11.   In this setting of recurrence risk evaluation, it’s important to consider adjuvant treatment. Authors should consider to reperform analysis with this variable

12.   Authors should consider to insert a limitation paragraph in order to discuss the small sample size of validation cohort, the selection of follow up time range for non-recurrent patients, and in general which are the limitations of DL method in this context and adding some considerations about the clinical application of the GAN and DL network to the patient’s risk classification.

Author Response

Reviewer 1: 

Open Review

Comments and Suggestions for Authors

Starting from the assumption that there are no standard clinic-pathological factors considered accurate in predicting recurrence of DCIS and that new findings highlight possible contribution of AI to investigate cancer prognosis prediction, the authors developed a GAN augmented deep learning (DL) classification framework using haematoxylin and eosin (H&E) images for predicting recurrence in DCIS patients. The DL model was mainly used to classify patients’ risk based on the differences identified between aggressive and non-aggressive images. They also studied the relationship between GAN classification and classical clinic-pathological parameters. The purpose of the study is original.

Thank you for your comments. We have tried to address all your concerns.

  1. Acronyms aren’t coherent with their order in text (for example H&E at line 56): they appear in text first without explanation. In general, there’s no explanation of BCE acronym.

Thank you for pointing out the issues. We have cross-checked our acronyms now.

  1. The quality of exposure is not always precise and there are some typos and sloppy mistakes. Some sentences are often not clear and too long. Grammar should be reviewed as the verb tenses are often contrasting.

Thank you for your comments. We have extensively edited the document and fixed the typos and grammatical errors.

  1. Author should report identification code of the protocol and institutional approval of their study

As archival de-identified samples were used in the study, a Waiver of Consent was given by Indiana University Institutional research Board. This is (now) clearly stated in the manuscript.

  1. Authors should explain better which kind of recurrence they have considered: local, loco-regional, ipsilateral…

As the cohort sizes are small, we used any “additional breast cancer event” whether local, contra-lateral or distant as an event. We will seek to stratify these in future studies in larger cohorts.

  1. How many mastectomies/lumpectomies in the cohort? What about resection margins in conservative surgery? Which margins was considered as free from tumour?

In a prior study using this cohort, we did not observe any differences based on the type of surgery or any other therapy (Figure 2 in Badve et al Br J Cancer 2021). However, given the small sizes of the cohort, we have not dissected the utility of L-GAN based on the types of treatment.

A statement that reads as follows has been included in the manuscript.

“Patients from both the cohorts were treated with mastectomy and/or a combination of lumpectomy, radiation and hormone therapy”.

  1. Did authors consider distant metastases too? If not, why not?

To the best of our knowledge, none of the patients with DCIS developed distant metastasis.

  1. Authors referred to a publication (reference n°22) for cohorts’ description but some questions have to be clarified:
  • Singapore cohort presented n=73 patients in reference n°22 but in this manuscript patients from the same cohort are 65. Authors should explain the reason of the exclusion and fill a table in text with baseline distribution of the new cohorts.

We excluded 3 patients due to missing follow up time (as included in ref 22) and 4 additional cases were excluded due to short follow up time (less than 3 years). So, the total sample size included in N=66, and we added Table 1 for detailed information.

  • Authors wrote “All cases had to have either a history of development of a second breast cancer event or a minimum of 3 years without any additional breast event” but from Supplementary Table S1 of reference n°22, follow up time (years) of non-recurrent patients starts respectively from 0.03 and 0.9 years for Oxford and Singapore cohort. Authors should explain why they included patients without a recurrence history and without a minimum of 3 years without any additional breast event. On the other hand, if they weren’t to be included, authors should consider reperform analysis with the corrected sample.
  • Authors should explain why they consider only a minimum of 3 years without recurrence: why not 5 years for example? Some references need to be addressed for this choice, this issue could be a selection bias. How long was the follow-up period the authors considered for recurrence diagnosis? 

Any event within 6 months was considered as residual disease and only classified as having BCEs. The 3-year limitation was essentially to prevent the loss of a substantial number of non-BCE patients. This has been added as a limitation of the study.

  1. In material and method section there isn’t any part related to how researchers collected H&E images, their quality, their dimension, their format, if they were elaborated before they were given as input to L-GAN, which part was considered... Author should add this part to the section.

Thank you for your comment. We have introduced image format, dimension and pre-processing that was performed in Section 2.1 now.

“The H&E image of each TMA core contains 7000 x 6000 pixels. A 2-class Gaussian mixture modeling was applied to each of the cores to separate tissue class vs background. A high-quality TMA core (i.e., no tears, intact) was manually chosen as a reference template. All other cores were normalized to the reference template using histogram equalization with 128 bins across RGB color channels. Patches with 256 x 256 pixels were extracted from each tissue core post histogram equalization for DL training and validation. Patches with less than 50% of area with tissue were automatically discarded from analysis.”

  1. Some of the basic notions about DL and GAN should be better explained, in order to be understood by a wider audience. For example: latent feature space, cosine similarity.

We have addressed the issues in Figure 1 and Section 2.2 now. We have explained how the latent feature space was created,

“P-GAN however is restricted to image generation and does not allow prognosis prediction. Feature space of P-GAN created from image patches from H&E image shows pathologically meaningful representation of lethal cancers clustered together in the embedded feature space.”

“The DL encoder/mapping network (Inceptionv3) within the GAN framework was re-tuned using a transfer learning strategy to learn the latent space of recurrent DCIS patches that comprised of embeddings of color, shape, texture and spatial features. The re-tuned encoding network was then used to extract features of the GAN generated image patches.”

We explained cosine similarity with reference and how it was used in our framework as,

“The generated image patches similar to original aggressive image patches in feature space were automatically selected based on cosine similarity {PMID: 30416537} or cosine angular similarity of the feature vectors. Cosine similarity of feature vectors extracted by the encoding/mapping network (from aggressive cancer patches and GAN generated image patches) were used to identify aggressive image patches generated by GAN.”

  1. Where was reported Table 1?

The error is regretted, and Table 1 is added to the manuscript.

  1. In this setting of recurrence risk evaluation, it’s important to consider adjuvant treatment. Authors should consider to reperform analysis with this variable

We are in complete agreement with the reviewer regarding the importance of the treatment variables. In our prior work (Badve et al Br J Cancer 2021), we had considered the treatment variables and that these did not impact the likelihood of BCEs in these cohorts. For this reason, they were not considered in the current analysis.

  1. Authors should consider to insert a limitation paragraph in order to discuss the small sample size of validation cohort, the selection of follow up time range for non-recurrent patients, and in general which are the limitations of DL method in this context and adding some considerations about the clinical application of the GAN and DL network to the patient’s risk classification.

We have introduced limitations of our method now in the Discussion section as below:

Though the framework for DCIS BCE prediction was validated in a holdout dataset there still some limitations to the study. The overall cohorts are small and the patients were treated in a non-homogenous manner. Surgical treatment in the form of mastectomy was routinely used in the Singapore cohort while lumpectomy was most often used in the Oxford cohort. Furthermore, there was almost no usage of endocrine therapy in the Singapore cohort. Due to the older nature of the Oxford cohort, endocrine therapy had been prescribed in only a small percentage of patients. BCE was defined as any breast cancer related event including local and distant disease; this was in part due to the small cohort sizes and low incidence of events in patients with DCIS. In the cohorts used, the follow-up non-BCE patients was limited resulting in the use of a 3-year endpoint. Although the cohorts are small, the multi-center nature of the study is major strength. Further validations of L-GAN are necessary prior to large-scale clinical implementation.

Reviewer 2 Report

The authors report a deep-learning method to predict breast DCIS recurrence. They used a pathology generative adversarial network (GAN) and beyond this a data augmentation method, called L-GAN. They got a good prognostic tool with AUC of 0.82, sensitivity of 0.81 and specificity of 0.83 in the hold out validation dataset. It seems very useful to evade unnecessary and potentially harmful treatment.

I am not fimiliar with this statistical method, since I am a clinician, however, I have some questions or remarks.

1. In the abstract the main result is "AUC of 0.82, sensitivity of 0.81 and specificity of 0.83 in the hold out validation dataset" which I couldn't find with certainity in the result section.

2. I think it would be worth emphasizing that the KM curve is about the Singapore dataset. Perhaps, it would be nice to see the result of the same analysis on the whole dataset.

3. Some acronym is unfolded several times (DL, GAN), but TNBC not even once.

Author Response

Reviewer 2: 

Open Review

Comments and Suggestions for Authors

The authors report a deep-learning method to predict breast DCIS recurrence. They used a pathology generative adversarial network (GAN) and beyond this a data augmentation method, called L-GAN. They got a good prognostic tool with AUC of 0.82, sensitivity of 0.81 and specificity of 0.83 in the hold out validation dataset. It seems very useful to evade unnecessary and potentially harmful treatment. I am not familiar with this statistical method, since I am a clinician, however, I have some questions or remarks.

Thank you for your encouraging remarks. We have tried to address all your concerns.

  1. In the abstract the main result is "AUC of 0.82, sensitivity of 0.81 and specificity of 0.83 in the hold out validation dataset" which I couldn't find with certainty in the result section.

Error is regretted. The Results section has been modified and the data is added.

“The re-tuned Inception-Resnetv2 classification model was validated using the Singapore cohort (n=66). Application of the model to the validation cohort resulted in a mean AUC of 0.82 for predicting BCEs with a sensitivity of 0.81 and specificity of 0.83.”

  1. I think it would be worth emphasizing that the KM curve is about the Singapore dataset. Perhaps, it would be nice to see the result of the same analysis on the whole dataset.

We emphasized that the KM curve is about the Singapore data set in the legend. KM curve on whole dataset is not included as we did not train Bayesian network (integrated model to create KM curve) on Oxford cohort to avoid overfitting as GAN and TILs were trained on Oxford cohort.

  1. Some acronym is unfolded several times (DL, GAN), but TNBC not even once.

Thank you for pointing out the issue, the error is regretted. We have cross-checked all acronyms now.

Reviewer 3 Report

In this work, the authors have developed a deep learning (DL) classification framework that can predict recurrence in DCIS patients from H&E images using a generative adversarial network.  A Bayesian network was also in place to integrate various modalities into interpretable personalized risk scores of recurrence.

There are a few concerns:

Major:

1. the results and discussion of this manuscript are shaped for data scientists entirely. There is a lack of orientation and presenting the clinical impact.

Minor:

2. summary and abstract are giving repeated info: "Standard clinicopathological factors (age,..." repeated 3 times.

3. the development key method of L-GAN (adapted from P-GAN) is not clearly presented. The readers have to go through references carefully to understand P-GAN first, and L-GAN. Figure 1 needs more instructions for showing the modification to develop the L-GAN framework.

4. same issue for Inception-Resnetv2 and Inception-Resnetv3 as point 2. 

5. There are clearly limitations to this study, especially the ones driven by data and DL nature, which are not discussed at all.

6. Code could be shared with the manuscript to support open science.

Author Response

 Reviewer 3: 

Open Review

Comments and Suggestions for Authors

In this work, the authors have developed a deep learning (DL) classification framework that can predict recurrence in DCIS patients from H&E images using a generative adversarial network.  A Bayesian network was also in place to integrate various modalities into interpretable personalized risk scores of recurrence.

There are a few concerns:

Major:

  1. the results and discussion of this manuscript are shaped for data scientists entirely. There is a lack of orientation and presenting the clinical impact.

We agree with your observations, one of the reasons for this was this article is to be included in a special issue on Artificial Intelligence. Furthermore, the primary function of this manuscript is to document the NOVEL functionality of GAN; this tool has never been used for predictive/ prognostic purposes. Lastly, as acknowledged, the small size of the cohort and the exploratory nature of the analysis are significant limitations to broad scale application. We in contact with UK/ANZ investigators and planning to perform a larger clinical study. However, without documentation of the feasibility of using the tool, access to the cohort is difficult.

We have added the following to the conclusion

“The current study is a proof-of-principle study that establishes the potential of using GAN models for prediction. The further studies are necessary to assess the impact of these predictive models on clinical practice.”

Minor:

  1. summary and abstract are giving repeated info: "Standard clinicopathological factors (age,..." repeated 3 times.

We regret the error.  The summary and abstract are modified (as below) to address the issue,

Simple Summary: Ductal carcinoma in situ (DCIS) patients have an excellent overall survival rate and over-treatment is always a cause for concern due to potential side-effects. Standard clinicopathological parameters have limited value in predicting breast cancer events (BCEs) and stratification of high and low risk patients. Herein, we have developed a deep learning (DL) classification framework to predict BCEs in DCIS patients. A generative adversarial network (GAN) augmented deep learning (DL) classification of histological features associated with aggressive disease was trained on hematoxylin and eosin (H&E) tissue microarray (TMA) images of DCIS to predict BCEs. The area under the curve (AUC) for BCE’s in the validation set was 0.82. Early and accurate prediction of DCIS BCEs would facilitate a personalized approach to therapy. 

Abstract: Standard clinicopathological parameters (age, growth pattern, tumor size, margin status and grade) have been shown to have limited value in predicting any Breast Cancer Event (BCEs) in ductal carcinoma in situ (DCIS) patients. Early and accurate BCE prediction would facilitate a more aggressive treatment policy for high-risk patients (mastectomy or adjuvant radiation therapy), and simultaneously reduce over-treatment of low-risk patients. Generative adversarial networks (GAN) are a class of DL models in which two adversarial neural networks, generator and discriminator, compete with each other to generate high quality images. In this work, we have developed a deep learning (DL) classification network that predicts breast cancer events (BCEs) in DCIS patients using hematoxylin and eosin (H&E) images. The DL classification model was trained on 67 patients using image patches from the actual DCIS cores and GAN generated image patches to predict breast cancer events (BCEs). The hold-out validation dataset (n= 66) had an AUC of 0.82. Bayesian analysis further confirmed the independence of the model from classical clinico-pathological parameters. DL models of H&E images may be used as a risk stratification strategy for DCIS patients to personalize therapy.

  1. the development key method of L-GAN (adapted from P-GAN) is not clearly presented. The readers have to go through references carefully to understand P-GAN first, and L-GAN. Figure 1 needs more instructions for showing the modification to develop the L-GAN framework.

We have addressed these issues now. In Section 2.2 to differentiate P-GAN and L-GAN we write,

“While the P-GAN[25] network is capable of generating high fidelity image patches with key tissue features such as color, texture, shape and spatial arrangement of both normal and cancer cells,  our L-GAN framework is specifically capable of capturing the aggressive image patches generated using the underlying P-GAN[25]. The DL encoder/mapping network (Inceptionv3) within the GAN framework was re-tuned using a transfer learning strategy to learn the latent feature space of recurrent DCIS patches that comprised of embeddings of color, shape, texture and spatial features. The re-tuned encoding network was then used to extract features of the GAN generated image patches. The generated image patches similar to original aggressive image patches in feature space were automatically selected based on cosine similarity {PMID: 30416537} or cosine angular similarity of the feature vectors. Cosine similarity of feature vectors extracted by the encoding/mapping network from aggressive cancer patches and GAN generated image patches was used to identify aggressive image patches generated by GAN. The selected aggressive image patches generated by GAN were then used to augment data for the subsequent DL classification model (Inception-Resnetv2) training for DCIS BCE prediction.”

We have also modified Figure 1 to better explain the framework.

  1. same issue for Inception-Resnetv2 and Inception-Resnetv3 as point 2. 

We have addressed this issue now in Section 2.1. Please check in comment 3.

  1. There are clearly limitations to this study, especially the ones driven by data and DL nature, which are not discussed at all.

We have addressed the DL and data driven limitations in our Discussion now,

“Though the framework for DCIS BCE prediction was validated in a holdout dataset there are certain limitations for the study. The overall cohorts are small and the patients were treated in a non-homogenous manner. Surgical treatment in the form of mastectomy was routinely used in the Singapore cohort, while lumpectomy was most often used in the Oxford cohort. Furthermore, there was almost no usage of endocrine therapy in the Singapore cohort. Due to the older nature of the Oxford cohort, endocrine therapy had been prescribed in only a small percentage of patients. BCE was defined as any breast cancer related event including local and distant disease; this was in part due to the small cohort sizes and low incidence of events in patients with DCIS. The cohorts used were enriched for BCE events and follow-up, particularly for the non-BCE patients was limited resulting in the use of a 3-year endpoint. The size limitations of the dataset for training may affect variability observed during training. This can adversely affect L-GAN image patch generation abilities and classification network performance downstream. Further validations of L-GAN are necessary prior to large-scale clinical implementation.”

  1. Code could be shared with the manuscript to support open science.

We will be happy to share the datasets and the code upon request with appropriate material transfer agreements.

Round 2

Reviewer 1 Report

The answers are not exhaustive. Authors didn't provide answer to one of our major question:

  • Authors wrote “All cases had to have either a history of development of a second breast cancer event or a minimum of 3 years without any additional breast event” but from Supplementary Table S1 of reference n°22, follow up time (years) of non-recurrent patients starts respectively from 0.03 and 0.9 years for Oxford and Singapore cohort. Authors should explain why they included patients without a recurrence history and without a minimum of 3 years without any additional breast event. On the other hand, if they weren’t to be included, authors should consider reperform analysis with the corrected sample.

Author Response

Comments and Suggestions for Authors

The answers are not exhaustive. Authors didn't provide answer to one of our major question:

  • Authors wrote “All cases had to have either a history of development of a second breast cancer event or a minimum of 3 years without any additional breast event” but from Supplementary Table S1 of reference n°22, follow up time (years) of non-recurrent patients starts respectively from 0.03 and 0.9 years for Oxford and Singapore cohort. Authors should explain why they included patients without a recurrence history and without a minimum of 3 years without any additional breast event. On the other hand, if they weren’t to be included, authors should consider reperform analysis with the corrected sample.

Response to Reviewer 1: We thank the reviewer for the careful attention to our earlier paper and we were remiss in not clearly explaining that for the current study. We excluded 2 cases from the Oxford cohort and 4 cases from the Singaporean cohort due to short follow up time (less than 3 years follow up for non-BCE/non-recurrence). To avoid confusion, we have removed the reference to the earlier study and now include Table 1 (also copied below) which includes the final number of patients included in the current study and demographic/clinical information. All analysis for the paper was conducted on these patients and thus, it is not necessary to repeat the analysis.
